# Consumers’ Knowledge and Handling Practices Associated with Fresh-Cut Produce in the United States

**DOI:** 10.3390/foods11142167

**Published:** 2022-07-21

**Authors:** Heyao Yu, Zhihong Lin, Michael S. Lin, Jack A. Neal, Sujata A. Sirsat

**Affiliations:** 1School of Hospitality Management, Pennsylvania State University, Mateer Building, University Park, PA 16802, USA; hvy5095@psu.edu; 2Conrad N. Hilton College of Global Hospitality Leadership, University of Houston, 4450 University Dr., Houston, TX 77204, USA; andrea.lin0126@gmail.com (Z.L.); jneal@central.uh.edu (J.A.N.); 3School of Hotel and Tourism Management, Hong Kong Polytechnic University, 17 Science Museum Rd, Tsim Sha Tsui, Hong Kong; michael.lin@polyu.edu.hk

**Keywords:** food safety assessment, food safety education, foodborne illness risk perception, demographic/generation factors, hierarchical linear regression

## Abstract

Previous studies have shown that three factors influence fresh-cut produce safety from farm to fork: (1) post-harvest practices in processing facilities, (2) employees’ handling practices in retail facilities, and (3) consumers’ handling practices in domestic kitchens or cooking facilities. However, few studies have examined consumers’ food safety knowledge, risk perceptions, and their handling practices associated with fresh-cut produce. To fill this gap, the present study conducted a nationwide survey to assess U.S. consumers’ food safety knowledge, practices, and risk perception associated with fresh-cut produce among various demographic groups and investigated factors influencing consumers’ food safety practices related to fresh-cut produce. The results showed that consumers lack the knowledge and safe handling practices toward fresh-cut produce regarding storage hierarchy, surface cleaning and sanitizing, and time and temperature control of fresh-cut produce. The men and millennial consumers exhibit a lower level of safe fresh-cut produce handling practices. In addition, a significant interaction was observed between food safety knowledge and risk perceptions on consumers’ fresh-cut produce handling practices, such that food safety knowledge can transfer to practice more effectively for consumers with high levels of risk perception. The results can be utilized to design effective consumer food safety education tools for targeted audiences.

## 1. Introduction

The World Health Organization (WHO) recommends that an adult should eat at least 400 g of produce every day to mitigate the risk of chronic illness and ensure an adequate daily intake of dietary fiber [1]. Therefore, many countries encourage fresh produce consumption through campaigns at the national government level. For example, in the United States (U.S.), the Department of Agriculture launched the MyPlate program [2] and, in the United Kingdom (U.K.), the Department of Health & Social Care [3] launched a program called the “Change4Life programme” to encourage people to eat more fresh produce and promote weight control. Moreover, consumers prepare less food at home due to a busier lifestyle and perceive fresh-cut produce as being healthy [4]. These government-led activities, the scientific literature, and consumer needs result in an increasing demand for fresh produce [5]. For example, the volume sales and dollar sales of fresh produce rose by 8% and 22% from 2010 to 2014 in the U.S. [6]. 

Within the fresh produce segment, the fresh-cut produce industry has developed rapidly in the last decade [7] and has become a substantial segment of the produce industry with multi-billion-dollar sales [8]. Based on the Statista report, the consumption of fresh-cut produce in the U.S. accounts for approximately 29.5% of fresh produce consumption [9]. The International Fresh-Cut Produce Association defined fresh-cut produce as “any fresh fruit or vegetable that has been physically altered from its original form but remains in a fresh state” [10]. According to the data from Supermarket Perimeter, fresh-cut and value-added vegetables, as a category, had a 6.6% increase in dollar sales in 2018 compared with 2017, partly due to convenience in terms of minimizing food preparation time [11,12]. Value-added vegetables include dehydrated, jammed, and pickled vegetables, which account for a relatively small portion of processed vegetables [13].

There has been a correlation between the increased intake of fresh and fresh-cut produce and the number of foodborne disease outbreaks associated with fresh and fresh-cut produce [4,14,15,16]. According to the U.S. Centers for Disease Control and Prevention (CDC) National Outbreak Reporting System [17], a total of 47,961 outbreaks were reported, where fresh produce accounts for approximately 60% of the outbreaks (n = 28,686) in the U.S. between 2000 and 2018. In addition, a rising concern from consumers exists about the risks associated with fresh-cut produce [18]. In particular, the frequency of foodborne disease outbreaks linked with fresh-cut produce has increased, despite the fresh-cut produce being perceived as safe by consumers [18,19,20]. A previous study found that among 606 produce-associated foodborne illness outbreaks, 73.3% of the outbreaks were associated with fresh-cut produce [21]. Most recently, a *Salmonella* Javiana outbreak in February 2020 related to fresh-cut mixed fruit (melons and pineapple) resulted in a total of 165 people infected across 14 states [22]. 

Fresh-cut produce has a higher risk of being associated with foodborne pathogen contamination due to additional processing procedures, including washing, cutting, and packaging when compared to other fresh intact produce [23]. Two possible reasons can explain why fresh-cut produce may carry higher foodborne illness risks compared with fresh produce [21]. First, processing of the produce (i.e., cutting, peeling, and shredding) can demolish cell surfaces and expose cytoplasm, which may offer another nutrient source for microorganisms than intact produce [24]. Second, most fresh-cut produce is consumed raw, meaning cooking is not used to inactivate and reduce pathogen loads [14]. Fresh-cut produce items that are ready to eat can make them carriers of foodborne pathogens if such items are not prepared and handled properly [16,18]. Previous studies have found that bagged fresh-cut romaine lettuce can contain equal levels of foodborne pathogens compared to whole leaf lettuce [15,25,26].

Three essential factors that influence fresh-cut produce safety practices in the farm to fork continuum were pointed out by food safety scholars: (1) post-harvest practices in processing facilities, (2) employees’ handling practices in retail facilities, and (3) consumers’ handling practices in domestic kitchens [27,28]. Previous studies have mainly focused on the first two factors: post-harvest and employees’ handling practices in retail facilities [4,15,29,30]. The current study focuses on consumers’ handling practices to alleviate foodborne illness risks. Additionally, although significant gender and generation differences in food safety knowledge and perception have been observed in the contexts of farmers’ market and foodservice operations [31,32], their impacts on food-safety-related knowledge and practices in the context of fresh-cut produce need to be examined. 

The overarching objectives of the current study were to investigate the knowledge, risk perceptions, and practices (self-reported) regarding fresh-cut produce among U.S. consumers and to determine the relationships among these factors. The results obtained from this study can be used to make targeted consumer food safety tools. Specifically, the present study conducts a national questionnaire to examine (1) U.S. consumers’ food safety knowledge, risk perceptions, and handling practices associated with fresh-cut produce; (2) the effects of demographic factors (i.e., gender and generation) on consumers’ food safety knowledge, risk perceptions, and food safety practices toward fresh-cut produce; and (3) the interaction effect of consumers’ risk perceptions on the relationship between food safety knowledge and food safety practices. 

## 2. Materials and Methods

### 2.1. Participants

A self-administered questionnaire was used to assess consumers’ food safety knowledge, handling, and risk perception associated with fresh-cut produce in domestic kitchens. Questions were selected from the previous literature and approved by the Human Subjects Review Board (IRB # 8171). A pilot test with 50 participants was conducted via an online panel (Qualtrics, Provo, UT, USA). Qualtrics can reach out to more than five million customer samples worldwide through its data collection collaborators to provide a sample that reflects the target population. A random sampling method was used in the pilot test because the purpose of the pilot test was to evaluate the clarity and reliability of the questions and responses. The questionnaire was modified according to the results of the pilot test, including providing more detailed information regarding the fresh-cut produce and simplifying the names of foodborne pathogens. The final questionnaire was also distributed nationwide via Qualtrics and data were collected during 2017–2018 (Qualtrics, Provo, UT, USA).

The questionnaire was equally distributed to 10 regions of the U.S. These regions were determined by the U.S. Environmental Protection Agency (EPA) with 105 respondents (10.1%) per region (Figure 1) [33]. Such a distribution method assured a reflection of a national representation of consumers’ knowledge, risk perceptions, and practices as it relates to food safety associated with fresh-cut produce in domestic kitchens. An initial screening question was included in the questionnaire to ensure that the respondents who were willing to participate in the study had purchased fresh-cut produce at least once a month on average. Moreover, three additional filter questions were included in the questionnaire to assure the quality of responses. A total of 106 responses were removed due to failure to answer the filter questions correctly, and 937 valid responses were used for the later data analysis. 

### 2.2. Measure Development

The questionnaire included three major sections. The first section consisted of questions assessing respondents’ food safety knowledge and practices regarding fresh-cut produce handling in domestic kitchens. Seven multiple-choice questions were adapted from the safe produce page on the Fight BAC! Partnership for Food Safety Education [34] to measure five fresh-cut produce handling practices and two general hygiene practices based on the recommended handling practices. The Fight BAC! campaign was launched in 1997 as a collaboration between the U.S. Food and Drug Administration (FDA), the CDC, and several leading food safety institutions, such as Institute of Food Technologists and International Fresh-cut Produce Association, aiming to help consumers reduce food safety risks. On the basis of the Fight BAC! Campaign’s recommendation, five practices specifically related to fresh-cut produce handling, include checking the “use-by” date while purchasing, storing fresh-cut produce above raw meat and poultry, storing fresh-cut produce at 41 °F or lower, disposing fresh-cut produce held at the room temperature for more than 4 h, and throwing away fresh-cut produce after passing the expiration date. These five produce handling practices have been validated by multiple independent studies [35,36]. The respondent was given “1” if he or she answered the question correctly, and “0” otherwise. An example question was “In your house, where is bagged salad/fresh-cut fruit stored in the refrigerator?” Upon completion of the knowledge assessment, seven items corresponding to the knowledge assessment questions were used to measure the frequency that respondents self-reported proper fresh-cut produce handling practices. The measure was rated on a five-point Likert scale (1 = never, 5 = always) with never = 0% of the time, occasionally = 20–30% of the time, sometimes = 50% of the time, frequently = 70% of the time, and always = 100% of the time [37]. An example item was “I store bagged salad or fresh-cut fruits and produce on the highest shelf in the refrigeration.” The correct answer to this question is “Top shelf”. 

The second section consisted of questions measuring the respondents’ risk perceptions toward developing a foodborne illness from consuming fresh-cut produce. Risk perception is defined as consumers’ perceptions of unexpected or poor consequences due to certain behavior [38,39]. In the paper that reviewed conceptualizations and models of consumer-perceived risks, Mitchell proposed several criteria to judge the validity of three existing perceived risk models, including (1) the level of concept understanding generated, (2) predictive success, (3) suitability for reliability testing, (4) level of practical and managerial insight offered, and (5) usability [40]. The normative expectancy-value (E-V) method, which views risk perceptions as a product of “probability and negative consequences”, received the highest score (3 out of 3) for all five criteria. Therefore, the present study adopted the E-V method to measure consumers’ perceived risks associated with fresh-cut produce food safety. Multiple studies conducted by the U.S. Department of Agriculture, CDC, and WHO consistently indicate that *E. coli* O157: H7, *Salmonella* spp., and *Listeria monocytogenes* are the most common pathogens related to fresh-cut produce [16,41,42,43]. Therefore, the respondents were asked to rate risks associated with *E. coli* O157:H7, *Salmonella* spp., and *Listeria monocytogenes* in fresh-cut produce. 

Five-point Likert scales were used to measure the actual risk (1 = “very safe”, 5 = “very risky”). An example item was “If there is *Salmonella* spp. in my fresh-cut produce, I think the implication for my health is ___.” Respondents were asked to answer the question “How likely do you think you encounter the following food-related issues when purchasing bagged salad and fresh-cut fruits and vegetables?” to assess the likelihood of food safety risks. Five-point Likert scales were used to measure the likelihood of this risk (1 = “very unlikely,” 5 = “very likely”). An example item was: “*Salmonella* spp. can be in the fresh-cut produce (fruits and vegetables) I purchase____.”.

The last section in this questionnaire included social demographic questions, such as questions about gender, age, education, income, and purchase frequency. Participants were categorized into three generations based on their self-reported age. Social scientists [44,45] contend that there are three generations in the most recent American history: the baby boomers (born between 1946 and 1964), generation X (born between 1965 and 1980), and millennials (born between 1981 and 1996). Recently, a new generation (Gen Z) is defined as individuals born between 1997 and 2010 [46] and our data showed that there were only five participants born between 1997 and 1999. Since the sample size was too small for any statistical analysis and they were among the oldest Gen Z population, they were included as millennial consumers in the present study. 

### 2.3. Data Analysis

The data were analyzed in three steps. The first analysis was descriptive analyses of the respondents’ demographic profiles and food safety knowledge and practices associated with fresh-cut produce in domestic kitchens. On the basis of a previous, consumers’ risk perception toward foodborne pathogens associated with fresh-cut produce was assessed as a single-order factor that reflected the multiplicative product of the level of risk toward a certain pathogen weighted by the likelihood with which the given risk was held [18]. To reduce the potential influence of a normality violation, a square root transformation was applied to the product. 

The second step demonstrated the differences in food safety knowledge and practices regarding fresh-cut produce among different demographic groups of consumers (i.e., gender and generation) using a multivariate analysis of variance (MANOVA) by IBM SPSS Statistics 25.0 for Windows (IBM Corporation, Armonk, NY, USA). In addition, hierarchical multiple linear regressions using ordinal least square estimation were used to examine food safety knowledge, risk perceptions, and the interaction effect of food safety knowledge and risk perceptions on fresh-cut produce handling practices, with the demographic factors as control variables. Bootstrapping analysis using the PROCESS add-on in SPSS (www.processmacro.org (accessed 1 July 2022)) was conducted to assure the robustness of the regression analyses. 

## 3. Results

### 3.1. Demographics

Table 1 shows the descriptions of the demographic attributes of the respondents. More than 60% of the respondents identified as women (61.8%, N = 579). The respondents included 511 millennials (54.5%), 224 baby boomers (23.9%), and 202 members of generation X (21.6%). Nearly 40% of the respondents received college degrees or above. Moreover, approximately 50% of the participants had an annual income between USD 35,000 and 79,999 (49.6%, N = 465). Approximately 48% of the respondents purchased and consumed fresh-cut produce two to three times weekly. 

A principal component analysis using Varimax rotation was conducted to explore the validity of the measures of food safety practices and food safety risk concerns. The results revealed a two-factor solution accounting for 72.2% of the overall variance. The eigenvalues of the two dominant factors were 4.85 and 2.37, respectively, indicating that the validity of food safety practices and risk concerns is acceptable. All of the factor loadings of food safety practice items on the first factor were above 0.7 and the factor loadings of risk perception items on the second factor were also above 0.7, indicating good measurement validity [47]. 

### 3.2. Consumers’ Knowledge and Practices Associated with Fresh-Cut Produce

Table 2 demonstrates the results of food safety knowledge associated with fresh-cut produce. Only 9.5% (N = 89) of the respondents correctly answered that bagged salad/fresh-cut fruit should be stored above raw meat and poultry in the refrigerator. In addition, 27.0% (N = 253) of the respondents correctly answered the question regarding kitchen surface cleaning and sanitizing, and only 34.9% (N = 327) of the respondents chose to throw away the bagged salad/fresh-cut fruits with no bruises or damage but passed the expiration date. 

In terms of food safety practices (Table 3), adapting the scoring system from previous studies [37,48], insufficient food safety practices related to fresh-cut produce are defined as any practice of which the average score is less than 4 (frequently, about 70% of the time). Respondents fell short in the following areas: storing bagged salad/fresh-cut fruits and vegetables on the highest shelf in the refrigerator (mean = 2.28 out of 5), putting a thermometer in my refrigerator to check the temperature (mean = 2.60), throwing bagged salad/fresh-cut fruits and vegetables away if they have been at room temperature for four hours (mean = 3.26), and throwing bagged salad/fresh-cut fruits and vegetables away if they pass the expiration date (mean = 3.56). The finding of food safety practices was consistent with that of food safety knowledge, indicating that consumers do not know or follow the food safety guidance regarding storage and time and temperature controls of fresh-cut produce [35,36,49].

### 3.3. Consumers’ Food Safety Knowledge and Practices Associated with Fresh-Cut Produce among Demographic Groups

MANOVA was used to compare the differences among social demographic factors, including gender, generation, and their interaction effect on consumers’ food safety knowledge and practices associated with fresh-cut produce. The results of the MANOVA shown in Table 4 indicated that the overall model was significant in terms of Wilks’ Lambda test (Wilks’ Lambda gender (2, 929) = 10.05, *p* < 0.01; Wilks’ Lambda generation (4, 1858) = 49.54, *p* < 0.01; Wilks’ Lambda gender*generation (4, 1858) = 10.34, *p* < 0.01). 

Consumers’ food safety knowledge (F_knowledge_ (1, 928) = 10.03, *p* < 0.01, ηp^2^ = 0.01) and food safety practices (F_practice_ (1, 928) = 4.23, *p* < 0.05, ηp^2^ = 0.01) associated with fresh-cut produce were significantly different between the men and women respondents. Table 5 shows the results of the Bonferroni post hoc test. The women (mean = 4.07) had higher food safety knowledge than the men (mean = 3.75) at a 0.01 statistical significance level. 

Significant differences also exist in consumers’ food safety knowledge (F_knowledge_ (2,928) = 45.09, *p* < 0.01, ηp^2^ = 0.09) and food safety practices (F_practice_ (2, 928) = 27.91, *p* < 0.01, ηp^2^ = 0.06) associated with fresh-cut produce among generations. The results of the Bonferroni post hoc test showed that the baby boomer respondents (mean = 3.31) had lower food safety knowledge than the generation X respondents (mean = 4.08) and the millennial respondents (mean = 4.34) at a 0.01 statistical significance level. However, the baby boomer respondents (mean = 3.95) exhibited higher food safety practice levels than the generation X respondents (mean = 3.69) and the millennial respondents (mean = 3.58) at a 0.01 statistical significance level. 

Additionally, a significant interaction effect was seen between gender and generation on consumers’ fresh-cut produce handling practices (F_practice_ (2, 928) = 14.82, *p* < 0.01, ηp^2^ = 0.03). An L-matrix analysis was used to further examine differences in consumers’ fresh-cut produce handling practices among six demographic groups. The results showed that baby boomer men (mean = 4.10) had significantly higher levels of food safety practices than generation X men (mean = 3.78) and millennial men (mean = 3.48). In addition, baby boomer women (mean = 3.80) had significantly higher levels of food safety practices than generation X women (mean = 3.60).

### 3.4. Consumers’ Food Safety Knowledge and Risk Perceptions Associated with Consumers’ Food Safety Practices

A hierarchical multiple regression analysis was conducted to examine the effects of food safety knowledge and risk perception and their interaction effect on consumers’ food safety practices. Model 1 examined the effects of control variables, including gender, age, income, and education on consumers’ fresh-cut produce handling practices. The result of Model 1, shown in Table 6, indicated that age (β_age_ = −0.01, *p* < 0.05) and education (_βeducation_ = −0.07, *p* < 0.05) significantly affected consumers’ food safety practices associated with fresh-cut produce. 

Model 2 examined the main effects of food safety knowledge and risk perceptions on consumers’ fresh-cut produce after controlling the effects of social demographic factors. After including food safety risk perceptions and knowledge in the regression model, consumers’ education level was not significantly related to their food safety practices (β_education_ = −0.03, *p* > 0.05). The result of Model 2 indicated that when controlling the effects of age, gender, income, and education, food safety knowledge (β_knowledge_ = 0.17, *p* < 0.05) positively affected consumers’ food safety practices associated with fresh-cut produce significantly. Particularly, in addition to the findings in Model 1, consumers who had more food safety knowledge implemented significantly safer fresh-cut produce handling practices than those who had less knowledge. However, consumers’ risk perception of foodborne illness was not significantly related to their fresh-cut produce handling practices (β_risk_ = 0.03, *p* > 0.05). 

Model 3 examined the interaction effect of food safety knowledge and risk perceptions on consumers’ fresh-cut produce handling practices. Except for age (β_age_ = −0.01, *p* < 0.05), other social demographic factors were not significantly related to consumers’ fresh-cut produce handling practices in Model 3. The result of Model 3 indicated that food safety knowledge (β_knowledge_ = 0.16, *p* < 0.05) and the interaction effect (β_interaction_ = 0.07, *p* < 0.05) positively affected consumers’ food safety practices associated with fresh-cut produce significantly. Bootstrapping analysis was conducted to assure the robustness of the regression analyses [50]. The result of the bootstrapping analysis was consistent with multiple linear regression models. The interaction effect between food safety knowledge and risk perceptions had a significant and positive effect on consumers’ food safety practices (β_interaction_ = 0.07, *p* < 0.05, 95% CI: [0.03, 0.09]). As shown in Figure 2, the result of the interaction effect suggested that the relationship between consumers’ food safety knowledge and fresh-cut produce handling practices was stronger for consumers with high levels of risk perceptions toward fresh-cut produce, compared with those with low risk perceptions.

## 4. Discussion

The present study investigated U.S. consumers’ food safety knowledge and handling practices associated with fresh-cut produce among demographic groups (i.e., gender and generation) and examined the interaction effect of food safety knowledge and risk perceptions on consumers’ fresh-cut produce handling practices. 

### 4.1. Consumers’ Food Safety Knowledge and Handling Practices Associated with Fresh-Cut Produce

The results showed that not enough consumers had food safety knowledge toward fresh-cut produce regarding safe storage order, surface cleaning and sanitizing, and time and temperature control of fresh-cut produce. Less than 10% of the respondents had knowledge regarding safe storage order of fresh-cut produce, and most of the respondents reported not storing their produce on the highest shelf in the refrigerator. This result demonstrated a significant food safety concern considering the criticality of placing ready-to-eat (RTE) food such as fresh-cut produce above raw poultry and meat (e.g., beef, lamb, and pork) to avoid potential cross-contamination according to the U.S. Food and Drug Administration Food Code [51]. Fight BAC! also suggests that consumer should store their ready-to-eat food above any raw meat and poultry in the consumer food safety education material “Top 10 Refrigerator Myth Fact” [52]. Safe storage order is critical to prevent juices or other liquids from raw poultry or meat products from contaminating food that will be served without cooking procedures [51]. The results are consistent with previous studies [53], which found that less than 40% of consumers were aware of the storage order of their refrigerators. 

In addition, less than 40% of the respondents reported that they disposed of fresh-cut produce with no bruises or damage after the expiration date. An independent study conducted in Spain [54] showed that approximately 10% of the consumers did not take into account the “use-by” date of fresh-cut produce. In the current study, more than 50% of the consumers reported disposing of fresh-cut produce if bruises or damages were visible. The result implied that consumers depended on the appearance of fresh-cut produce to determine the food safety status, which is consistent with a previous study which indicated that consumers consider appearance as a primary criterion of the acceptability of fresh produce [7]. As suggested by the USDA Food Safety and Inspection Service (FSIS) [55], the “use-by” date should not be used as an indicator of safety but rather of food quality. However, several studies have shown that RTE food with an initial contamination of 5 CFU/g *Listeria monocytogenes* could reach a level of more than 100 CFU/g at the “use-by” date even when RTE food is held at 41 °F [56,57]. 

### 4.2. Consumers’ Fresh-Cut Produce Knowledge and Handling Practices among Gender and Generations

The MANOVA results showed that gender and generation influenced consumers’ food safety knowledge and handling as it relates to fresh-cut produce significantly. More specifically, women exhibited higher levels of food safety knowledge and enhanced handling practices (self-reported) associated with fresh-cut produce compared to men. Moreover, baby boomer consumers exhibited significantly lower levels of food-safety-related knowledge compared to generation X and millennial consumers, whereas baby boomer consumers demonstrated safer handling practices compared with younger generations. The findings of the effects of demographic factors on consumers’ food safety knowledge and practices associated with fresh-cut produce are consistent with previous studies [58,59,60,61]. An independent study [59] surveyed 475 college students and indicated that women demonstrate significantly greater food safety knowledge compared to men. Another study [61] conducted a national survey (n = 1108) regarding consumers’ food safety and quality perceptions and behaviors in Australia and the results showed that women exhibited safer handling practices and had more food-safety-related concerns as compared to men. 

### 4.3. Factors Associated with Consumers’ Fresh-Cut Produce Handling Practices

The last component of the study examined the interaction effect between consumers’ risk perception and food safety knowledge associated with fresh-cut produce on consumers’ handling practices. The results showed that food safety knowledge was significantly related to consumers’ fresh-cut produce handling practices, whereas consumers’ food safety risk perceptions related to fresh-cut produce did not influence fresh-cut produce handling significantly. In addition, the interaction effect between food safety knowledge and food safety risk perceptions related to fresh-cut produce on consumers’ handling practices was significant. Most previous studies examined the effects of food safety knowledge and risk perceptions on consumers’ safe food handling practices separately [62,63,64]. Multiple consumer behavior studies have indicated that risk perception can also be an important situational factor that moderates the relationship between information stimulus and consumers’ evaluation and perceptions [65,66]. The results showed that the relationship between food safety knowledge associated with fresh-cut produce and consumers’ fresh-cut produce handling practices was stronger when consumers had higher levels of risk perceptions.

## 5. Conclusions

The study results demonstrated that a large proportion of consumers lack food safety knowledge regarding fresh-cut produce, especially in terms of safe storage order and time and temperature control of fresh-cut produce. In addition, the results revealed that few consumers follow the instruction of the expiration date on the package and instead depend on produce appearance as a measure of safety. The demographic factors had a significant influence on consumers’ food safety knowledge and handling practices associated with fresh-cut produce. This study has several limitations. First, FDA [67] also suggests that “never allow raw meat, poultry, seafood, eggs, or produce that requires refrigeration to sit at room temperature for more than 2 h; the limit is one hour if the air temperature is above 90 °F”. However, because our knowledge assessment was developed based on the Fight Bac! safe produce page, we did not examine consumers’ fresh-cut produce handling practices when the temperature is above 90 °F. Future research can test consumers’ knowledge regarding food handling practices when the temperature is above 90 °F. Second, the self-reported measures of fresh-cut produce handling practices may not accurately reflect consumers’ real handling practices compared with obtaining observational data. Therefore, a multi-method and multi-trait study, such as collecting self-reported and observational data, can be conducted in the future to provide a more robust measure of food handling practices. 

## 6. Practical Implications

The results of the present study provide several key recommendations for the fresh-cut produce industry and consumer food safety education. First, the results of consumers’ food safety knowledge and handling practices associated with fresh-cut produce suggest that consumers lack knowledge regarding safe storage order, cleaning and sanitizing the food-contact surface, and time and temperature control of fresh-cut produce. These results can guide public food safety education initiatives. In addition, public education programs can utilize videos and posters in addition to booklets to enhance consumers’ knowledge related to safe storage order, time and temperature control, and cleaning and sanitizing, given that previous food safety studies suggested that visual training methods were more effective in terms of improving individuals’ knowledge level [68,69]. In addition, the results showed that women demonstrated higher levels of food safety knowledge and practices associated with fresh-cut produce than men. The results suggest that public food safety education programs should enhance men’s food safety knowledge and practices associated with fresh-cut produce. The current study results found that lower levels of risk perception toward fresh-cut produce might hinder the transfer of food safety knowledge to practices associated with fresh-cut produce. The results suggest that public food safety education materials should not only focus on food safety knowledge provision but also enhance consumers’ risk perceptions toward fresh-cut produce consumption. Previous literature suggested that personal relevance and fidelity of the education materials could significantly increase individuals’ risk awareness [70,71]. Thus, future food safety education related to fresh-cut produce provided by cooperative extension educators and produce industry can use a storytelling method [72] to provide real-life examples about foodborne disease outbreaks resulting from contaminated fresh-cut produce to enhance consumers’ risk awareness toward food safety.

## Figures and Tables

**Figure 1 foods-11-02167-f001:**
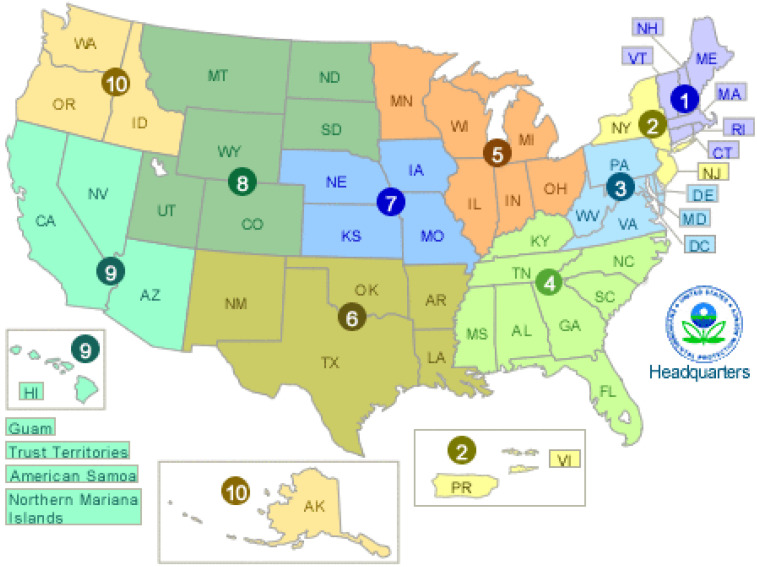
U.S. Environmental Protection Agency (EPA) region distribution. The numbers demonstrate ten regions within the U.S. A total of 105 respondents participated from each of the 10 regions (n = 937 valid responses).

**Figure 2 foods-11-02167-f002:**
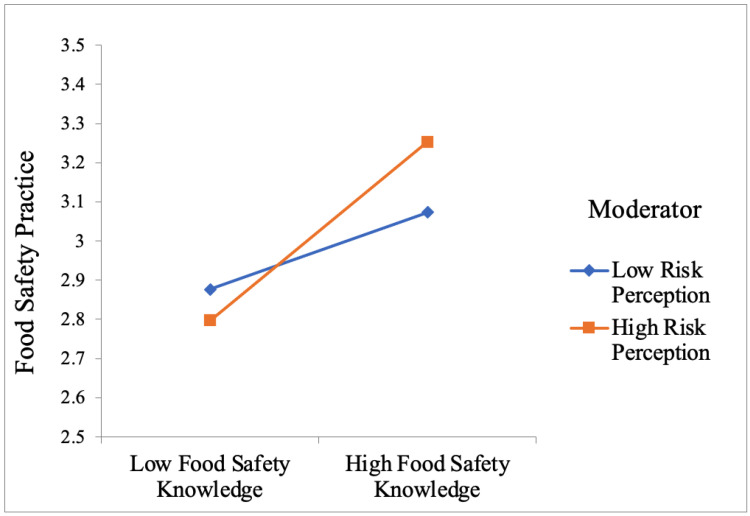
Interaction effect between food safety knowledge and risk perception on consumers’ fresh-cut produce handling practices.

**Table 1 foods-11-02167-t001:** Descriptive analysis of demographics of respondents (N = 937).

Demographic Factors		N (Percentage)
Gender		
	Male	358 (38.2%)
	Female	579 (61.8%)
Generation		
	Baby Boomers (above 60)	224 (23.9%)
	Generation X (36–59)	202 (21.6%)
	Millennials (18–36)	511 (54.5%)
Education		
	Basic (high school or equivalent)	565 (60.3%)
	Advanced (college degree or above)	372 (39.7%)
Income		
	Low (USD 34,999 or lower)	202 (21.6%)
	Middle (USD 35,000–79,999)	465 (49.6%)
	High (above USD 80,000)	270 (28.8%)
Purchasing frequency		
	Low (once per week)	257 (27.5%)
	Middle (two or three times per week)	450 (48.0%)
	High (more than three times per week)	230 (24.6%)

**Table 2 foods-11-02167-t002:** Descriptive analysis of food safety knowledge associated with fresh-cut produce.

Abbre.	Item	N	Correct Percentage
FSK1 ^a^	Checking “use by date” while purchasing	662	70.7%
FSK2	Handwashing before food preparation	778	83.0%
FSK3	Food-contact surface cleaning/sanitizing	253	27.0%
FSK4	Storing fresh-cut produce at 41 °F or lower	536	57.2%
FSK5	Storing fresh-cut produce above raw meat and poultry	89	9.5%
FSK6	Throwing away (leftover)	459	48.9%
FSK7	Throwing away (expiration)	327	34.9%

^a^ FSK: food safety knowledge associated with fresh-cut produce. N = 937.

**Table 3 foods-11-02167-t003:** Descriptive analysis of food safety practices associated with fresh-cut produce.

Abbre.	Item	Mean	S.D.
FSP1 ^a^	Checking “use by date” while purchasing	4.56	0.76
FSP2	Handwashing before food preparation	4.53	0.76
FSP3	Food-contact surface cleaning/sanitizing	4.20	0.94
FSP4	Storing fresh-cut produce at 41 °F or lower	2.60	1.61
FSP5	Storing fresh-cut produce above raw meat and poultry	2.28	1.36
FSP6	Throwing away (leftover)	3.26	1.36
FSP7	Throwing away (expiration)	3.56	1.24

^a^ FSP: food safety practices associated with fresh-cut produce. S.D. = standard deviation. N = 937.

**Table 4 foods-11-02167-t004:** Multivariate analysis of variance of food safety knowledge and practices by gender and generation.

Source	Dependent Variable	Type III Sum of Squares	df	F	Partial ηp^2^
Corrected Model	Knowledge	203.70	5	22.42 **	0.11
	Practice	29.73	5	16.03 **	0.08
Intercept	Knowledge	10,890.16	1	5992.56 **	0.87
	Practice	9967.76	1	26,879.59 **	0.97
Generation	Knowledge	163.89	2	45.09 **	0.09
	Practice	20.70	2	27.91 **	0.06
Gender	Knowledge	18.22	1	10.03 **	0.01
	Practice	1.57	1	4.23 *	0.01
Generation*Gender	Knowledge	7.35	2	2.02	0.004
	Practice	10.99	2	14.82 **	0.03

**F**_gender_ (2, 929) = 10.05, *p* < 0.001, **F**_generation_ (4, 1858) = 49.54, *p* < 0.001; **F**_gender*generation_ (4, 1858) = 10.34, *p* < 0.01 in Wilks’ Lambda test. * *p* < 0.05. ** *p*< 0.01.

**Table 5 foods-11-02167-t005:** Main effect of food safety knowledge and food safety practices.

Factor and Its Attribute Level			
**Food safety knowledge**	**Male**	**Female**	**Main effect means**
Baby Boomer	3.00	3.61	3.31 **
Generation X	4.02	4.14	4.08 **
Millennials	4.23	4.44	4.34 **
Main effect means	3.75 **	4.07 **	
**Food safety practices**	**Male**	**Female**	**Main effect means**
Baby Boomer	4.10	3.80	3.95 **
Generation X	3.78	3.60	3.69 **
Millennials	3.48	3.68	3.58 **
Main effect means	3.79 *	3.69 *	

* *p* < 0.05; ** *p* < 0.01.

**Table 6 foods-11-02167-t006:** Hierarchical multiple regression analysis with all potential variables.

Factors	Coefficient	t-Statistic	Sig.
Model 1 (F = 11.752; R^2^ = 0.05)			
Constant	4.23	29.27	0.00
Gender	−0.04	−0.89	0.37
Age	−0.01	6.05	0.00
Income	0.022	1.83	0.07
Education	−0.07	2.18	0.03
Model 2 (F = 36.477; R^2^ = 0.22)			
Constant	4.33	31.00	0.00
Gender	−0.08	−1.93	0.06
Age	−0.01	7.93	0.00
Income	0.02	1.98	0.06
Education	−0.03	1.01	0.31
Food safety knowledge	0.17	8.16	0.00
Risk Perception	0.03	1.46	0.15
Model 3 (F = 13.092; R^2^ = 0.30)			
Constant	4.31	30.95	0.00
Gender	−0.08	−1.80	0.07
Age	−0.01	7.91	0.00
Income	0.02	2.09	0.06
Education	0.03	1.06	0.29
Food safety knowledge	0.16	7.89	0.00
Risk Perception	0.03	1.27	0.21
Food safety knowledge × Risk Perception	0.07	3.62	0.00

## Data Availability

The data presented in this study are available on request from the corresponding author.

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
