# Peer review of "Consumers’ Knowledge and Handling Practices Associated with Fresh-Cut Produce in the United States"

_foods, 2022, doi:10.3390/foods11142167_

Round 1

Reviewer 1 Report

The relationship between factors like education and knowledge is well already established, and the study's primary objective is not mentioned.

What would be the outcome of the survey is not mentioned.

Line 63: Salmonella Javiana  Javiana is a strain, so should not be in italics as per scientific nomenclature

Line 220, Explain, what does eigen value predicts in the discussion part

The discussion section is too lengthy, it shall be crispy and shall not be repetitive.

Line 528-552: Redraft the paragraph in a maximum of 5-6 lines. The relevance of the study in terms of its future use may also be included in the conclusion part.

In the conclusion part, the authors mentioned a long list of limitations that may draw all attention of the readers. After reading the extensive discussion, as a reader, the question came across my mind, whether the study is a more or less repetitive one and if the limitations were already known, the experimental design could be improvised that way to address the same. Further, a plagiarism report generated from Turnitin indicates a 23% similarity.

Author Response

Thank you so much for your valuable feedback and comments. We have addressed each comment below and in the manuscript narrative.

Reviewer 1

The relationship between factors like education and knowledge is well already established, and the study's primary objective is not mentioned.

Response: Thank you for your comment. We have included the overarching objectives in the manuscript narrative. It was previously highlighted as research questions but these are objectives.

What would be the outcome of the survey is not mentioned.

Response: Thank you for your comments. The results of this study can be used to design targeted food safety tools for consumer’s. We have included this point in multiple sections within the manuscript narrative.

Line 63: Salmonella Javiana  Javiana is a strain, so should not be in italics as per scientific nomenclature

Response: Thank you for your comment. We have edited this in the manuscript narrative.

Line 220, Explain, what does eigen value predicts in the discussion part

Response: Thank you for your comment. We have addressed what these values mean in the manuscript narrative.

The discussion section is too lengthy, it shall be crispy and shall not be repetitive.

Response: Thank you for your comment. We appreciate this insight and agree with you. We have deleted several components of the discussion section and reviewed for flow in the manuscript narrative.

Line 528-552: Redraft the paragraph in a maximum of 5-6 lines. The relevance of the study in terms of its future use may also be included in the conclusion part.

Response: Thank you for your comment. We have addressed this section in the manuscript narrative.

In the conclusion part, the authors mentioned a long list of limitations that may draw all attention of the readers. After reading the extensive discussion, as a reader, the question came across my mind, whether the study is a more or less repetitive one and if the limitations were already known, the experimental design could be improvised that way to address the same.

Response: Thank you for your comment. We have addressed your comment in the manuscript narrative.

Further, a plagiarism report generated from Turnitin indicates a 23% similarity.

Response: Thank you for your comment. We went through the document side by side with the results uploaded and addressed this throughput the manuscript narrative.

Reviewer 2 Report

The manuscript deals with nationwide survey to assess U.S. consumers’ food safety knowledge, practices, and risk perception associated with fresh-cut produce among various demographic groups .

The manuscript is well written. It deals with a very interesting topic for a wide range of readers. It will be very valuable to the community.

Although the study covers consumers ages ranging from 18 years and above 60 years. But it neglected teenagers below 18 years who could be potential consumers for fresh-cut produce. In addition, in p.11 the authors discussed the effects of Listeria monocytogenes neglecting other very virulent microorganisms as Giardia lamblia, Hepatitis A, Noroviruses, Rotavirus, Shigella

Author Response

Reviewer 2

Thank you so much for your valuable feedback and comments. We have addressed each comment below and in the manuscript narrative.

The manuscript deals with nationwide survey to assess U.S. consumers’ food safety knowledge, practices, and risk perception associated with fresh-cut produce among various demographic groups. The manuscript is well written. It deals with a very interesting topic for a wide range of readers. It will be very valuable to the community.

Response: Thank you for your comments. We appreciate your time in reviewing our manuscript and providing feedback.

Although the study covers consumers ages ranging from 18 years and above 60 years. But it neglected teenagers below 18 years who could be potential consumers for fresh-cut produce.

Response: Thank you for your comments. We agree that this is a crucial demographic; however, the IRB was not designed to collect data from individuals below 18. This will be a good opportunity for a future follow up study though.

In addition, in p.11 the authors discussed the effects of Listeria monocytogenes neglecting other very virulent microorganisms as Giardia lamblia, Hepatitis A, Noroviruses, Rotavirus, Shigella

Response: Thank you for your comments. We agree with your comments. Based on another reviewers comments, we have deleted these sections in the manuscript discussion to make it more targeted and crisp.

Reviewer 3 Report

The manuscript titled “Consumers’ Knowledge and Handling Practices Associated with Fresh-Cut Produce in the United States " it is innovative  and presents interesting results.

I would like to thank you for the opportunity to review this manuscript. Although the topic is interesting, the manuscript needs some revisions.

In particular, the literature review could be enriched. Furthermore, the authors could evaluate the possibility of providing a specific paragraf for the literature review.

2.1 Participants 

In this paragraph the period in which the survey was carried out should be specified.

4.4 Practical Implications

This paragraph should be moved to the end, after the conclusions.

Author Response

Reviewer 3

Thank you so much for your valuable feedback and comments. We have addressed each comment below and in the manuscript narrative.

The manuscript titled “Consumers’ Knowledge and Handling Practices Associated with Fresh-Cut Produce in the United States " it is innovative  and presents interesting results.

Response: Thank you for your comments. We appreciate the positive feedback.

I would like to thank you for the opportunity to review this manuscript. Although the topic is interesting, the manuscript needs some revisions. In particular, the literature review could be enriched. Furthermore, the authors could evaluate the possibility of providing a specific paragraph for the literature review.

Response: Thank you for your comments and positive feedback. Based on your comments and comments from other reviewers, we have made edits and addressed the literature review and discussion section to read better and avoid repetitions.

2.1 Participants

In this paragraph the period in which the survey was carried out should be specified.

Response: Thank you for your comment. We have included the years data was collected in the participants section in methods.

4.4 Practical Implications

This paragraph should be moved to the end, after the conclusions.

Response: Thank you for your comment. We have moved the Practical Implications paragraph after conclusions.